# Clinical Viability of Boron Neutron Capture Therapy for Personalized Radiation Treatment

**DOI:** 10.3390/cancers14122865

**Published:** 2022-06-10

**Authors:** Dominika Skwierawska, José Antonio López-Valverde, Marcin Balcerzyk, Antonio Leal

**Affiliations:** 1Department of Biomedical Engineering, Faculty of Electronics, Telecommunications, and Informatics, Gdańsk University of Technology, 80-233 Gdańsk, Poland; dominika.skwierawska@gmail.com; 2Departamento de Fisiología Médica y Biofísica, Facultad de Medicina, Universidad de Sevilla, 41009 Seville, Spain; jlvalverde@us.es; 3Instituto de Biomedicina de Sevilla, IBiS, 41013 Seville, Spain; 4Unidad Ciclotrón, Centro Nacional de Aceleradores, Universidad de Sevilla-CSIC-Junta de Andalucía, 41093 Seville, Spain

**Keywords:** BNCT, targeted therapy, biological dosimetry, boron imaging, personalized oncology, personalized medicine

## Abstract

**Simple Summary:**

Usually, for dose planning in radiotherapy, the tumor is delimited as a volume on the image of the patient together with other clinical considerations based on populational evidence. However, the same prescription dose can provide different results, depending on the patient. Unfortunately, the biological aspects of the tumor are hardly considered in dose planning. Boron Neutron Capture Radiotherapy enables targeted treatment by incorporating boron-10 at the cellular level and irradiating with neutrons of a certain energy so that they produce nuclear reactions locally and almost exclusively damage the tumor cell. This technique is not new, but modern neutron generators and more efficient boron carriers have reactivated the clinical interest of this technique in the pursuit of more precise treatments. In this work, we review the latest technological facilities and future possibilities for the clinical implementation of BNCT and for turning it into a personalized therapy.

**Abstract:**

Boron Neutron Capture Therapy (BNCT) is a promising binary disease-targeted therapy, as neutrons preferentially kill cells labeled with boron (^10^B), which makes it a precision medicine treatment modality that provides a therapeutic effect exclusively on patient-specific tumor spread. Contrary to what is usual in radiotherapy, BNCT proposes cell-tailored treatment planning rather than to the tumor mass. The success of BNCT depends mainly on the sufficient spatial biodistribution of ^10^B located around or within neoplastic cells to produce a high-dose gradient between the tumor and healthy tissue. However, it is not yet possible to precisely determine the concentration of ^10^B in a specific tissue in real-time using non-invasive methods. Critical issues remain to be resolved if BNCT is to become a valuable, minimally invasive, and efficient treatment. In addition, functional imaging technologies, such as PET, can be applied to determine biological information that can be used for the combined-modality radiotherapy protocol for each specific patient. Regardless, not only imaging methods but also proteomics and gene expression methods will facilitate BNCT becoming a modality of personalized medicine. This work provides an overview of the fundamental principles, recent advances, and future directions of BNCT as cell-targeted cancer therapy for personalized radiation treatment.

## 1. Introduction

Cancer, a multicellular and multigenic disease, is one of the leading causes of death in the world. It is the first/second cause in 112 out of 183 countries and the third/fourth in 23 countries, according to estimated data from the World Health Organization (WHO) in 2019. In 2020, 19.3 million new cases and 10 million cancer deaths were recorded according to GLOBOCAN estimates of 36 cancers in 185 countries [1]. Given that the annual incidence continues to increase, clinical management of cancer remains a significant challenge. Cancer can arise from all organs and different cell types with a multifactorial etiology. In general, cancer cells exhibit inherent phenotypical characteristics, known as hallmarks of cancer. Hanahan and Weinberg [2] originally suggested six alterations in cell physiology that collectively dictate malignant growth: environmental independence for growth, evasion of apoptosis (programmed cell death), limitless proliferative potential, sustained angiogenesis, tissue invasion, and metastasis to other parts of the body. In a more recent update, they also included deregulated metabolism and immune system evasion as additional hallmarks, as well as two characteristics that allow the acquisition of all hallmarks: genome instability and inflammation [3].

In countries with a high gross domestic product, radiotherapy (RT) is used in more than 50% of patients to treat the disease in a local stage or to control and alleviate symptoms of irrecoverable cases, depending on the stage of cancer [1]. Currently, estimates of the global demand for RT by cancer patients indicate that, for 87% of new cases of breast cancer patients, RT is an important part of therapy according to clinical guidelines and evidence-based medicine. In the case of surgery in the early stages of breast cancer and systemic therapies in metastatic cases, RT is applied as an adjunctive therapy [4]. In several types of tumors, such as head and neck cancers, skin cancers, cervical cancers, brain tumors, and others, RT is considered the alternative curative treatment option or even the basis for definitive curative treatment. Additionally, in many locations of the disease, RT is usually required as a prophylactic agent after surgery. In locations such as the lungs, the precision of the radiation beams allows the use of RT under ablative conditions when surgery is not applicable. This is the case of Stereotactic Ablative Radiotherapy (SABR), where vascular endothelial injury and immune activation are new radiobiological aspects that must be added to the 5 R’s of radiobiology (reoxygenation, repair, radiosensitivity, redistribution, and repopulation) to explain the ablation effect.

These data indicate that the role of RT has increased in prominence compared to surgery in the treatment of localized solid tumors, the most widespread expression of cancer. Among other reasons, these achievements are the result of several innovative therapeutic methods and technological improvements, such as the implementation of devices to shape and adapt the irradiation beam to the tumor volume while safeguarding organs at risk, or the successful implementation of advanced imaging procedures used in planning and treatment [5,6]. RT aims to deliver the optimal dose to tumor volume while preserving normal tissues, but these volumes are considered at the macroscopic level in treatment planning following an evidence-based population medicine approach through the image processing for segmentation of the target volume in the image data of patients usually provided by CT devices [7]. Unfortunately, patients often vary between tumor responses to RT due to differences in tumor type and other specific genetic factors not yet considered in treatment planning.

Beyond technological or clinical constraints, the efficacy of RT is strongly limited by the different biological characteristics of the tumor. Personalized medicine is usually related to genetic or other biomarker information to make treatment decisions for individually considered patients. Under this definition, RT cannot play a leading role in personalized cancer treatment as it has been played so far by chemotherapy and immunotherapy. However, efficient synergies between methodologies with a basic scientific profile and other more technical ones that have revolutionized RT could be the key to the real clinical implementation of personalized cancer therapy. In this sense, Boron Neutron Capture Therapy (BNCT) shows interesting possibilities insofar as it provides cellular targeting of radiation energy transfer and so could make more efficient previous genetic and molecular information about each patient.

BNCT is a binary treatment method based on the combination of two agents, ^10^B and epithermal neutrons, that exploits the high linear energy transfer (LET) characteristics of the mixture of fractionation components. Boron can be selectively localized in tumor cells. Thus, BNCT is a promising disease-targeted therapy, as neutrons preferentially kill cells labeled with ^10^B, making it a treatment modality of precision medicine. The overall BNCT procedure, indicating the principal steps from diagnosis confirmation to post-treatment monitoring, is shown in Figure 1.

Two types of neutron beam are commonly used in BNCT: thermal beams (~0.0254 eV) and epithermal beams (0.5 eV to 40 keV). For clinical purposes, the most useful are epithermal neutrons because, while entering the tissue, they create a radiation field with maximum thermal flux at a depth of 2–3 cm, which then drops exponentially. In turn, when a thermal beam enters the tissue, the thermal flux—which is created as a result—falls exponentially from the surface.

A fundamental principle of BNCT method is ^10^B(n,α)^7^Li nuclear reaction, which occurs when the stable isotope ^10^B, which is administered preferentially to tumor cells, is subsequently irradiated with an external epithermal neutron beam to produce an α-particle (^4^He) and a ^7^Li nucleus. A schematic representation of this reaction is presented in Figure 2. Released α-particles (~1.47 MeV) and the ^7^Li nuclei (~0.84 MeV) have a high LET ~175 keV·μm^−1^ [8,9,10]. About 94% of the time, the recoiling ^7^Li ion is produced in an excited state and emits a low LET 477 keV gamma-ray during deexcitation. In the remaining 6% of events, ^7^Li is emitted with no gamma-ray emission in the ground state.

Unfortunately, after many years of research led by scientists and specialists, they are still struggling with some critical issues of BNCT. Firstly, generating a therapeutic beam with an optimal energy spectrum that can deliver the neutrons to the correct location while minimizing the dose delivered to healthy tissue. Second, finding non-toxic ^10^B delivery agents: the boron carrier compound is one of the fundamental aspects of BNCT and should bring significantly more ^10^B to the cancerous cells than to the healthy tissue or, ideally, only into the cancerous cells. It should also meet requirements, such as water solubility, chemical stability, and preservation of constant high concentration during the treatment procedure [11]. Because the boron concentration level directly affects the yield of alpha and Li particles generated and hence the dose to the tumor and other tissue, it is also essential to image the boron distribution when considering BNCT. Several modalities are widely used to assess the boron dose delivered to the residual tumor volume, and they can provide information on the distribution of ^10^B at the microscopic level [12,13]. To improve molecular imaging, several other approaches have been proposed [14,15]. Another critical issue is developing treatment planning programs and systems to calculate the dose, predict particle fluxes, and expect the incidence angles in patients to achieve an adequate relative dose distribution. If the reactor generates the BNCT treatment beams, the treatment planner must determine the dose induced by neutrons and gamma photons. In this case, most of the gamma photons that occur in the beam originate from the reactor core. Fortunately, development research on compact, in-hospital accelerator-based neutron sources, ready for installation in hospital environments, has been ongoing for many years in several countries. It allowed for more widespread use of the BNCT technique and removed the contamination from high energy gammas from the reactor core, which was one of the reasons for the failure of BNCT in the 1950s.

Numerous reasons define why the cell may die. They relate to the cell cycle phase in which irradiation occurs, the cell type, the radiation dose, and physiological conditions, such as oxygen supply. However, what drives cancer cells during the switch from a repair program to cell death and what drives the cancerous cell to choose a specific pathway of death? The specific mechanism of cell death and the mechanism of repair after BNCT are not sufficiently known. In this case, the evaluation of early and late markers of cellular responses after the introduction of BNCT should be considered. They are crucial for the further development of BNCT.

BNCT could be treated as a personalized treatment, as ^10^B uptake depends on factors such as the characteristics of the individual tumor cell, the pharmacokinetics of boron drugs, and its subcellular distribution. Moreover, the optimal moment of boron-carrying drug concentration ratio in tumor cells vs. healthy ones, which varies from patient to patient, must be achieved. This review highlights the issues described above and evaluates them in future directions and the further development of BNCT as an effective personalized radiation therapy.

The answers to the questions presented and others that will undoubtedly arise in the further development of research on cancer radiation therapy, and the progression in solving the issues presented, even in part, will eventually lead to continued improvement in BNCT-based cancer treatment.

## 2. Fundamental Aspects of BNCT

BNCT is primarily a biochemically, rather than physically targeted type of radiation therapy, so fundamental aspects are first introduced in this section, such as the characteristics of BNCT as a targeted and binary therapy, the possible mechanisms and types of BNCT-induced cell death that occur during and after irradiation, and to pay attention to the necessary biological dosimetry to understand and evaluate the increased radio-sensitization effect.

### 2.1. BNCT as Targeted Radiotherapy

BNCT could enable highly individualized tumor treatment in the sense that the therapeutic effect can be limited to the specific tumor spread of the patient. In BNCT, this will only be possible when a compound accumulates ^10^B only in all tumor cells with a sufficiently high concentration in relation to healthy cells. Nuclear localization is preferred to maximize DNA damage. Therefore, boron delivery agents are one of the essential aspects of BNCT. ^10^B should be retained in the tumor, at least for the duration of neutron irradiation, which can take up to an hour. However, the way to concentrate ^10^B in sufficient amounts and preferentially in cancer cells is currently the main limitation of the effectiveness of BNCT. The first step is the separation of ^10^B (20% abundance) from ^11^B (80%), which is mastered but expensive process. Many compounds have been developed to date, but currently only two boron agents are widely used as boron carriers: sodium borocaptate (Na_2_B_12_H_11_SH [[^10^B]BSH]) and [^10^B]4-borono-L-phenylalanine (BPA), two drugs containing low molecular weight boron.

BSH used in BNCT consists of 12 ^10^B atoms and is used mainly to treat malignant gliomas. BSH is not delivered to the normal brain through the intact blood–brain barrier (BBB), and it is difficult to selectively internalize in tumor cells because of its high hydrophilicity. Its concentration in the target is related to the concentration of the agent in the blood and the vascularization of the neoplasm [16]. BSH has a passive diffuse accumulation mechanism. In malignant cells in the brain, it accumulates only in the tumor region where the blood–brain barrier is disrupted [17].

On the other hand, BPA is a derivative of phenylalanine and is actively transported into tumor cells, mainly through the L-type amino acid transporter 1 (LAT 1) [18]. The BPA logP is negative (−3.65), and this indicates that it will not pass the BBB passively because only small molecules with logP in the range of 1.5–2.7 can cross the BBB with passive (diffusive) transport [19,20]. Therefore, the only way for BPA to cross the BBB is to sneak it through some transporter, such as an L-type amino acid. BPA has been reported to accumulate specifically in tumor cells due to its structural similarities to tyrosine [5]. As L-type amino acid transporters are involved in some important human diseases and are overexpressed in human tumors, they improve targeted delivery to the brain and cancer cells. LAT 1 is also present in the blood–brain barrier (BBB), the blood–retina barrier, the cerebral cortex, testes, placenta, and bone marrow. Injection of [^10^B]BPA for intravenous administration in BNCT is prepared as the [^10^B]BPA-fructose complex [21]. The reason for labeling these compounds with positron emitters is to accurately determine the boron distribution and concentration in the tumor and surrounding tissue using PET, as will be discussed in Section 3. BPA was approved in Japan as a commercial drug with social security reimbursement and has been available on the market since 20 May 2020 under the name Borofalan (^10^B) [22].

Three generations of boron compounds can be distinguished: (I) borax, boric acid, and its derivatives used in the first clinical trials [23,24]; (II) boron-modified amino acids, including boron carriers, such as BPA and BSH; and (III) a third generation of boron agents, which has attracted the attention of scientists over the past two decades. They focus on using biochemical pathways to accumulate boronated analogs in subcellular structures. These new BNCT agents include small molecules containing boron, such as peptides [25,26], antibody-based delivery systems [27], boron compound conjugates [28,29,30], boron-dispersed nanoparticles [31,32,33,34] (nanomaterial-based delivery systems), and others currently under evaluation. Targeted boron delivery agents combine boron-containing agents with tumor-targeting molecules (e.g., nucleosides, porphyrins, peptides, proteins, or antibodies). Boron-delivery nanomaterials can transport various boron-containing compounds into tumor cells by taking advantage of the enhanced permeability and retention effect that affects nanomaterials and the active targeting effects mediated by tumor-targeted ligands grafted on the surface of the materials. The following boron-delivery nanomaterials can be distinguished: dendrimers, liposomes, polymeric nanoparticles, boron nitride, carbon nanotubes, mesoporous silica nanoparticles, ferromagnetic and paramagnetic nanoparticles—some of them used for MRI imaging—, gold nanoparticles, and BPO_4_ nanoparticles. There are many promising routes in drug delivery systems, and there is still a pressing need to develop new boron delivery agents, but without adequate research and clinical trials, it is difficult to determine which is the most feasible [35,36].

There are some obstacles to the exploitation of the full potential of the cancer-specific selectivity of BNCT, including the suboptimal boron delivery strategies currently used. However, various supportive targeted measures are under development, for example, delivery agents targeting the glucose transporter GLUT1 [37].

In any case, BNCT workflow would allow for a full personalization of the treatment, from the drug delivery treatment to imaging, and considering high throughput techniques, as it is discussed later in the “BNCT and personalized therapy” section.

### 2.2. Mechanisms of Cell Death in BNCT

In BNCT, cells are damaged mainly by alpha particles or the ^7^Li ion, as they can cause various DNA lesions along their path, i.e., DNA damage in clusters or multiple local damage sites, resulting in genome instability. In addition to DNA, macromolecules can also be damaged, resulting in modulation of their functions [38]. When these particles pass through a cell, their path is short (α < 10 µm and ^7^Li < 5 µm) [39], so their kinetic energy is released within the target cell, whose diameter is usually ~10 µm. Therefore, it does not affect the surrounding healthy cells. Intracellular boron localization is critical because normal healthy tissue can be spared from nuclear reactions if it does not uptake ^10^B. Unfortunately, with the currently available boron carrier compounds, some ^10^B also accumulate in healthy cells. The development of boron carriers is still highly active today [40]. A schematic representation of the principles of action at cellular level is presented in Figure 3.

Radio-induced damage can be produced through two types of action: direct and indirect. In the first case, radiation directly affects DNA, causing the ionization of atoms within the DNA molecule in small fractions of seconds. However, radiation-caused ionization must take place within a few nanometers of the DNA molecule to directly damaging it. As with indirect damage, radiation interacts with other target molecules or atoms that it encounters, usually water [41]. As a result, highly reactive species, such as HO and H, are produced and they can diffuse some distance in the cell before reaching the place where damage will be produced.

Different types of DNA injury include base damage, DNA protein or DNA–DNA crosslinks, double-strand breaks (DSB), single-strand breaks (SSB), sugar-phosphate backbone interruption, etc. Their distribution and repair pathways depend strongly on the type of radiation used during BNCT and its LET characteristics [11,42].

DNA damage generally increases together with radiation LET [43], and the higher the LET, the higher the relative biological effectiveness (RBE). Takatsuji et al., discussed the relationship between LET and RBE considering chromosomal aberrations and cell death, and found that at low doses, RBE increases with increasing LET, then the RBE value peaks at a LET of about 100 keV·μm^−1^, and, finally, the RBE decreases as LET increases further [44]. The radiation field generated during BNCT consists of components with different LET characteristics that act independently. Low-LET radiation ionizes sparsely, whereas high-LET radiation causes denser ionization along the track and can lead to more complex DNA damage. DNA damage after high-LET radiation remains unrepaired for a long time, leading to genome instability or cell death [45]. The density of radiation affects the presence and quality of radiation induced DSB.

Like many cancer treatments, radiation therapy achieves its therapeutic effect by causing a reaction of several types of cell death: apoptosis, mitotic cell death or mitotic catastrophe, necrosis, autophagy, and others. Apoptosis or mitotic cell death are the most common types, along with necrosis. Radiation-induced apoptosis is a progressive and degrading process. Extrinsic (death receptor) and intrinsic (mitochondrial) apoptosis can be distinguished. Intrinsic apoptosis is a type of regulated cell death (RCD) that is initiated by perturbations of the intracellular microenvironment and marked by permeabilization of the mitochondrial outer membrane. The extrinsic apoptotic pathway is triggered by binding of death ligands to transmembrane receptor proteins (e.g., TNF-α to TNFR1). Mitotic cell death is a specific variant of RCD driven by mitotic catastrophe, an oncosuppressive mechanism to control mitosis-incompetent cells [46]. Wang et al. [47] confirmed that in glioma cells, BNCT-induced apoptosis was mediated by the Bcl-2/Bax pathway.

Another important goal of radiation therapy is to deprive cancer cells of their potential to divide and multiply indefinitely [7]. The primary and presumed cell target of the ionizing radiation is DNA itself. However, damage or mutations in different cellular macromolecules cannot be fully eliminated, and as a result, their functions could be modulated, and other subsequent biological changes can be observed after cancer treatment [38].

The DNA-DSB repair process is complex and depends on many factors, including the cell cycle phase and checkpoints, DSB-inducing agents, non-coding RNAs, and various gene mutations characterized by different cancer cell lines. Several attempts have been made over the past few years to investigate the specific response to cellular DNA damage induced by a mixed neutron-gamma field [45,48,49]. Despite that, this phenomenon is not fully understood and determined.

Rodriguez et al. [50] have attempted in vitro studies of DNA damage and repair mechanisms induced by BNCT. The human thyroid follicular cancer cell line was used for the research. DNA damage assessment was performed by detecting H2AX histone phosphorylation foci (γH2AX foci). Repair of DBS through two mediating pathways has been identified in mammalian cells, homologous recombination repair (HRR) and non-homologous end joining (NHEJ). After the analysis of the follicular carcinoma cell analysis, the repair pathways were observed with an increase in Rad51 and Rad54 mRNA expression 4 and 6 h after irradiation, showing the expression of enzymes that belong mainly to the HRR pathway, specifying a different pattern of DNA damage and showing activation of both repair pathways. However, what exactly determines the activation of HRR or NHEJ is not yet completely clear.

To increase anticancer biological activity during BNCT therapy, Ikuhiko Nakase et al., performed an in vitro BNCT assay [26]. The study also evaluated cell death pathways to understand the cell killing activity that occurs after thermal neutron irradiation. They synthesized and demonstrated organelle-targeted cell-penetrating peptide (CPP)-conjugated boron compounds. CPPs help to control intracellular localization, cell membrane penetration, and further enhance cellular uptake of the boron compound. This controlled delivery affects the types of cell death and the efficacy of cancer cell killing activity. Treatment with DB-RLA ((BODIPY)-labeled dodecaborates conjugated to an RLA peptide) showed a higher reduction in the ATP content than other peptides tested. ATP depletion enhances necrosis, which consequently might induce necrosis in BNCT. This could be one of the significant factors of the cell-killing activity and detailed mechanisms should be further studied.

### 2.3. Physical Basis and Dosimetry of BNCT

Ionizing radiation has many forms, from alpha, beta, proton, or neutron particles, to X or gamma rays, and others. The main components that contribute to the total absorbed dose rate in BNCT are the elastic interaction of incident neutrons with hydrogen, the gamma ray dose emitted by the source, and the thermal neutrons captured by hydrogen, nitrogen, and boron [51]. Each of the components has various biological weighting factors. The total biologically absorbed dose (Gy-Eq) is the sum of the physical dose components (D) multiplied by the compound biological effectiveness (CBE) or relative biological effectiveness (RBE) of each dose component. RBE is the ratio of the dose absorbed from the reference radiation to the value of the radiation dose tested, producing the same biological effect, whereas CBE represents values of the biological efficacy for each dose component, depending on the boron carrier used. CBE differs from RBE with high LET general radiation in that the value varies depending on target cells, tissue, and the type of boron compounds used [52]. However, because of the occurrence of events at the cellular and subcellular levels, the different energies and types of radiation involved, the dosimetry, and accurate estimation of RBE, CBE, and, therefore, the biological effectiveness of BNCT is challenging.

Streitmatter et al. [53] presented a multiscale system of dosimetric and radiobiological models that better assess biological effectiveness. It can predict not only CBE and RBE but also other critical biological metrics for neutron sources, such as boron micro-distribution and tissue types. The model was tested against results from published experiments in vitro and in vivo, with and without boron, and showed good agreement between both.

As the biodistribution of boron varies from patient to patient, determining the boron concentration, as will be discussed in Section 3, is one of the crucial factors often marked as a drawback in BNCT because it causes ambiguity in the calculated dose distributions. For effective treatment with BNCT, the ratio of ^10^B concentration in the tumor and its concentration in normal tissues (T/N ratio) should be 3:1 or more, and the concentration of ^10^B in the target should be at least ∼15–30 μg g^−1^ or ∼10^9^ atoms per cell to perform lethal tumor cell damage [54]. In summary, to avoid unfavorable effects, the concentration of ^10^B in tumor cells and normal tissues must be known to support the calculation of the total dose distribution and to allow a good prognosis by allowing appropriate patient selection and optimization based on the spatial distribution of the boron-containing compound known before treatment.

Human tissue also contains certain isotopes that react with neutrons. Due to the values of nuclear cross sections, the most meaningful interactions of neutrons with human tissue involve ^1^H, ^12^C, ^14^N, and ^16^O isotopes, which account for 99.2% of all atoms in the human body [55]. Consequently, all the described components should be considered during the evaluation and calculation of the doses received because they could also be responsible for the adverse effects of BNCT. The most advanced methods of calculating fluxes and doses in complex geometries with a heterogeneous physical density, such as those of the patient are based on Monte Carlo (MC) techniques. This mathematical method uses the probability distribution that describes the transport of particles to determine the outcome of each step of their history along the way through the mediums by randomly sampling. In this way, you can know the position, direction, and energy, among other variables, of each particle in the radiation fluxes. Today, it is accepted that the probability description of particle transport is accurate enough with the energy values involved, so the MC method is the gold standard tool for dose calculation or design of new prototypes for the generation of neutron sources.

Before a clinical application, BNCT, like any other radiotherapeutic technique, requires a preceding dose calculation to determine the proper operation of the device and the irradiation condition by considering all design features and beam shaping components with respect to material, motion, and geometry for dose delivery. These processes are carried out by the treatment planning systems (TPS) [56]. Despite the high time consuming inherent to the Monte Carlo method, this shows a high accuracy compared to the other analytic algorithms. Thus, most BNCT TPS use a MC method to estimate the absorbed dose in some part of the whole process. Many different tools and dose calculation algorithms based on the MC method have been developed and are being developed continuously that can assist clinicians in personalized treatment planning and decision making. Nowadays, the dose distribution and mean doses absorbed in regions of interest (ROI), and the dose volume histograms and isodose curves superimposed on personalized anatomical models of the patients can be extracted and displayed graphically after appropriate data calculation. The shorter the recording step to track the transport of particles along the material media described in the image, the more accurate MC dose calculation is, but a smaller registration volume for the calculation of the average dose value relates to higher statistical uncertainty due to a lower number of events in the volume. Therefore, the computation time is an important handicap for the clinical application of MC.

Some attempts have been made to improve the MC algorithms, e.g., by increasing the dose calculation speed of BNCT TPS, such as in GPU-accelerated Monte Carlo [57]. Although some companies are working on a commercial MC dose engine focused specifically on the clinical application of BNCT, the MC method is often used as an alternative tool to verify specific commercial TPS [58]. The general-purpose MC radiation transport code MCNP [59] has been used recurrently as a dose calculation engine in BNCT. Historically, the ‘Simulation Environment for Radiotherapy Application’ (SERA) [60] has been used in BNCT facilities using a special MC calculation engine. On the other hand, more recently, Hu et al., evaluated the Particle and Heavy Ion Transport code System (PHITS) [61], a multimodal Monte Carlo code developed by the Japan Atomic Energy Agency in Japan, for micro dosimetry in BNCT [62]. It can help to evaluate doses in radiobiological experiments. In addition, it can consider intracellular and intercellular heterogeneity in the ^10^B distribution. Therefore, it was proposed as a model that can estimate the biological effectiveness of newly developed ^10^B compounds for BNCT, which would be advantageous in future drug discovery research. The study resulted in the general conclusion that PHITS can be applied to evaluate the dose rates of absorbed gamma rays and the thermal neutron fluxes within a tumor-imitating medium [62,63].

In the clinical dose calculation, several components of the absorbed dose must be considered. Four main components of the absorbed dose in BNCT can be distinguished:Fast neutron dose: according to ^1^H(*n, n*)*p* reactions, fast and epithermal neutrons cause elastic neutron collisions with hydrogen in tissue (giving recoiling protons and gammas). Other energy depositions from fast neutron reactions like ^12^C(*n*, α) are also included.Incident and secondary gamma ray’s dose: Primary gamma dose from the beam port and secondary gamma dose by ^1^H (*n*, γ)^2^H. This component can be used for real time dosimetry with SPECT imaging, as it is further described in Section 3.Nitrogen dose: according to the ^14^N (*n*, *p*)^14^C reaction, the ^14^N element in the tissue captures a thermal neutron and, as a result, a ~600 keV proton is emitted. The dose is obtained from locally delivered energy from the recoiling ^14^C nucleus and the energetic proton.Boron dose: energy deposited by the ^10^B (*n*, α)^7^Li reaction. ^10^B captures a thermal neutron, and as a result, an alpha particle and a recoiling ^7^Li ion are emitted. The dose derived from the reaction products is ~2.31 MeV.

The cross sections for some of these reactions are shown in Figure 4.

In addition, the neutron capture reaction of Cd in single-photon emission CT detectors can be used for real time dosimetric purposes, as it is discussed in Section 3.3.

The recoil ionization of hydrogen is the leading way by which neutrons with energy >0.01 MeV deposit the dose. However, the ^14^N (*n, p*)^14^C reaction at neutron energies <1 eV is responsible for ~80% of the energy released in the tissue [55]. Overall, 88.8% of thermal neutrons are absorbed in the ^1^H (*n*, γ)^2^H reaction, and 10.6% of thermal neutrons are absorbed in the ^14^N (*n, p*)^14^C reactions. Additionally, in the reactions mentioned above, the ^14^N atom also loses an electron. However, the proton and electron do not combine instantly as ^1^H. For this, the proton is moving too rapidly through the tissue (Q = 0.58 MeV) and will cause further ionization due to the high LET. The emitted proton average residual range in soft tissue (after entering the high-LET Bragg peak phase) is longer than the diameter of a typical cell nucleus but shorter than the diameter of a typical human cell. It is also necessary to estimate the cell-killing potential of the ^14^N (*n, p*)^14^C reaction and consider it since adenosine of ATP, ADP, AMP, DNA, and RNA bases and other common molecules, such as NADH, contain a significant amount of nitrogen. In a tissue exposed to a dose arising from a fast neutron beam, cells killed by ^14^N (*n, p*)^14^C reactions compared to those killed by recoil proton and heavy-ion tracks are imperceptible [55]. Furthermore, the doses of ^14^C decay compared to background radiation and the statutory limits are not significantly lower. The fraction of respiratory phosphate molecules—i.e., AMP, ADP, ATP, NADH, etc.—that undergo the ^14^N (*n, p*)^14^C reaction are negligible at therapeutic neutron doses. Determining the dose resulting from reaction ^14^N (*n, p*)^14^C is necessary in situations where people may be exposed to prolonged exposure to significant thermal neutron fluxes [55].

As a result, the dosimetry of BNCT requires an in-depth analysis of various components of the radiation field. To predict a biological effect, the dose arising from each of these four components must first be multiplied by an appropriate biological weighting factor to account for differences in relative biological effectiveness and, ultimately, combined [65]. Accepted values of biological weighting factors are 1.3 for the dose of boron in normal tissues, 3.8 for the dose of boron in the tumor, 3.2 for the thermal and fast neutron dose, and 1 for the dose of gamma.

Beyond these weighting factors, although the calculation of the absorbed dose is a good starting point to discuss the possible biological impact of BNCT on RT, it is not enough to assess its complete potential radiobiological expected benefit. In this way, many efforts are being made to include the chemical and biological aspects of the problem in MC calculations, such as the code developed within the Geant4-DNA Project [66] (TOPAS-nBio is a wrapper for the latter). A clear example of this is the work by Perry et al. [67] to model the DNA damage produced by the high LET particles involved in BNCT. Anyway, these theoretical calculations must be compared with experimental verifications.

### 2.4. BNCT Biological Dosimetry

As noted previously, the effect of BNCT is highly dependent on a biological component. Thus, it is crucial to assess the promoted increased radio-sensitization effect, in addition to the physical dose enhancement. This is a hard task that depends specifically on in vivo or in vitro studies, which involve methodologies such as proliferation tests, clonogenic tests, or the evaluation of DNA damage.

Sung et al. [68] performed clonogenic tests, evaluating survival in terms of the proliferative capacity of irradiated cells, and obtained a dose-dependent suppression of cell survival when treated with BPA under BNCT irradiation schemes. This effect showed up to ~10 times less survival when boron was present during a ~3 Gy irradiation. Furthermore, they also analyzed the mitochondrial metabolic activity of irradiated cells with the 3-(4,5-dimethylthiazol-2-yl)-2,5-diphenyl tetrazolium bromide (MTT) assay. The results showed a significant decrease in metabolic activity in different cells irradiated with BPA compared to cells irradiated without BPA, ranging from ~20% up to ~80% at 3 days after irradiation, depending on the cell line evaluated. This result suggested a decrease in proliferative capacity after BNCT. In addition, they also pointed out cell cycle arrest at G2/M checkpoints and an increase in apoptotic cells after BNCT versus neutron irradiation, using flow cytometry assays. The increase in apoptotic cells and cell cycle arrest in G2/M was confirmed in terms of increased expression of caspase-9 and cytochrome c and decreased expression of cyclin B1 and CDK1, respectively, using Western blots. These results are consistent with reports from other studies [47,63]. Moreover, newer studies even proposed mathematical models that fit data from experiments that study the same cellular parameters related to biological effectiveness [69].

In any case, the radio-sensitization effect could be observed at a more precise level in terms of DNA damage. Therefore, studies that assess the presence of DSB repair markers, such as γH2AX foci, have provided further information on the matter. This is the case of the study by Rodriguez et al. [50] which determined that the number of localized lesions was lower when comparing gamma-ray radiation with neutron or BNCT radiation, but the damage caused by BNCT was densely concentrated in clusters, correlated with the expected more complex damage caused by high LET radiation. Moreover, these large foci lesions were persistent when observed for longer timeframes, describing firm or irreparable long-term damage [70,71]. Thus, despite an initial lower γH2AX foci count, the BNCT DNA damage profile involves more complex and irreparable damage patterns that would mean a higher radiobiological effect. In any case, further studies could be implemented using some of the automatic quantification algorithms for γH2AX foci [72,73] could be implemented for a more exhaustive and robust foci quantification.

Furthermore, it is well known that dose rate plays a crucial role in radio-sensitization [74]. Hence, the long-term effect of the appearance of such discrete events of large-dose deposition prompted by BNCT remains to be determined. These events differ greatly from the more continuous events that occur in conventional γ irradiation, depositing less dose each one.

At the same time, BNCT treatment has been shown to alter levels of cellular oxidative stress, both due to BNCT itself and due to tumor-targeting boron carriers [75,76]. The effects of these on oxidative stress changes in the biological effective dose and radio-sensitization need to be further studied.

## 3. Boron Analysis and Boron Imaging in BNCT

BNCT agents deliver boron atoms precisely to tumor cells, maintaining an appropriate concentration higher in the tumor than in normal tissue. The effectiveness of the therapy depends on where the neoplastic cell drug was in the neoplastic cell population and within the tumor cells. The intranuclear location of boron increases the chances of killing cells by DNA damage. The lack of a method for a quantitative imaging evaluation of the boron concentration was always one of the issues that nuclear doctors faced when using neutron irradiation. Therefore, methods for evaluating the three-dimensional distribution per patient of boron drugs, boron dose, and all complex radiation compositions delivered to residual tumor volume and healthy tissue are one of the most critical issues of BNCT. Chemical imaging of cellular and subcellular levels is necessary to support clinical efficacy, dosimetry studies, and general new drug delivery research in BNCT. To solve this problem and achieve selective tumor accumulation and reduced toxicity, several approaches have been applied, e.g., coating, functionalizing, labeling with different fluorophores or molecules with fluorescence properties [14,15].

The boron concentration level directly affects the intensity of the boron neutron capture reaction and the dose to the tumor and other tissue, as discussed previously. Therefore, it is essential to image the local boron concentration while considering BNCT in every patient treated with that therapy to calculate the delivered radiation dose and determine the optimal neutron irradiation time in a personalized manner. Consequently, alternative methods to predict blood boron levels must be developed and evaluated between measurements and during irradiation. A substantial improvement in BNCT will be achieved when the boron concentration is measured in situ. Furthermore, it should also be noted that the uptake of boron-carrying molecules in target cells is heterogeneous. It depends on factors, such as tumor cellularity (that is, the number of tumor cells arranged in clusters), cell cycle phase, and others [77,78].

### 3.1. Positron Emission Tomography and Magnetic Resonance Imaging

Clinically applicable imaging modalities are positron emission tomography (PET) [79] and magnetic resonance imaging (MRI) techniques (^1^H in BPA) [79]. PET is one of the key tools for imaging and studying biochemical in vivo processes in clinical use. PET has many abilities, such as (I) quantifying biochemical processes, (II) reconstructing the distribution of a boron carrier (this information can be later used in treatment planning), (III) finding and determining the extent of metastasis in the body, (IV) predicting the optimal time of neutron exposure in BNCT, (V) controlling the therapeutic effects, and (VI) assessing whether the patient is suitable for BNCT. The suitability of PET to establish boron concentration in healthy tissues and tumors and the need for treatment planning have been examined in many studies [77,80,81]. PET imaging with current technology can measure the boron distribution prior to treatment. As a result, the therapeutic dose distribution calculated with PET may disagree with the actual dose delivered. However, based on values such as the ability to estimate the concentration ratio of ^10^B in a tumor compared to adjacent normal tissues and determining treatment indications, it is possible to decide whether BNCT treatment will be beneficial for the patient. This, in turn, also allows to decide the patient’s eligibility for BNCT treatment and additionally enables high-precision personalized treatment planning. The most common radio-labelled derivative of BPA used to estimate BPA concentration in vivo through PET is [^18^F]FBPA (4-Borono-2-^18^F-fluoro-L-phenylalanine). PET using [^18^F]FBPA is a useful treatment strategy for BNCT and the technique of determination of the boron concentration in tumor and normal tissues based on [^18^F]FBPA molecular imaging has developed dynamically in recent years. Scientists are still conducting studies to detect a compound with greater potential for non-invasive quantification of local boron concentration by PET imaging, for example, the theranostic agent itself—metabolically stable boron-derived tyrosine [82].

One of the main tasks of personalized medicine is to consider the drug response based on the patient’s genotype, the gene expression profile, and other individual characteristics. As BNCT is a binary treatment, where in addition to neutron flux, the boron-containing drug is still in experimental choice—besides Borofalan, which, as mentioned earlier, is clinically approved. The evaluation of the interaction of the drug with the individual patient is still in its early stage [83,84]. The further complication is that [^18^F]FBPA and BPA differ by ^1^H-^18^F substitution, and also in that [^18^F]FBPA is administered in μg amounts, while BPA is in 300 mg/kg. BPA peak blood concentration is a few tens of times higher than phenylalanine. Although, in the future theranostic approach, the ^10^B-containing drug and the PET-imaged one may be the same, the required doses of the ^10^B-containing drug will need to exceed many times the baseline level of the drug analog. This is the case for the BPA–phenylalanine (or tyrosine) pair and still for glucose analogs, where this discrepancy will be smaller [37,85] as blood and tissue glucose levels are on the order of the required ^10^B concentration in tissue.

In the paper [86], Balcerzyk et al. explored the possibility of PET measurement of boron concentration if the compound contains the *R*-BF_3_ moiety that labels it with ^18^F. This method was applied to [^18^F]NaBF4 used in the preclinical study of thyroid cancer.

Some research evaluated the use of [^11^C]-methionine ([^11^C]Met)—the most popular radiolabeled amino acid that plays an important role in protein synthesis—as an alternative candidate to [^18^F]FBPA. In some of them, this method was reported to be used as a patient indicator for BNCT instead of [^18^F]FBPA PET for some types of cancer [87,88]. [^11^C]Met can be especially useful in facilities unable to synthesize [^18^F]BPA by themselves. It can also be used to evaluate ^10^B uptake in tumors in BNCT trials, such as in Yamamoto’s [89] comparative study (phase II BNCT study of glioblastoma) of [^18^F]FBPA and [^11^C]Met.

MRI is also used as a modality for indirect quantification of the in vivo distribution of boron at the target site, during and before neutron irradiation, making it suitable for BNCT [90]. There are fewer studies that applied MRI as an imaging technique, but this method has some advantages over [^18^F]FBPA PET, e.g., less invasiveness and more versatility with fewer restrictions of a boron–gadolinium compound. MRI also has an excellent spatial resolution for soft tissues, which is beneficial in some cases, for example, in head and neck tumors. It can also provide functional and morphological information without using radiation, which makes it safer for the patient and the observation time window is significantly larger as scan can be repeated without causing any toxic effects. For this purpose, to obtain high-contrast images, it is necessary to introduce non-toxic ^10^B molecular compounds tagged with a paramagnetic ion into the body, such as gadolinium, which will work as an MRI reporter during the mapping of the boron distribution [91,92,93,94,95]. Currently, agents that conjugate Gd and boron are in the phase of animal experimentation stage.

Measurement of the net content of ^10^B atoms, bound and free boron pools, and the factors that affect the content in individual tumor cells are not widely described in the literature and remain challenging, as PET and magnetic resonance modalities do not offer sufficient spatial resolution to quantify boron atoms in single cells [96]. Nevertheless, knowledge of the “micro-distribution” in the tumor of boron containing drugs may offer benefits of personalized tailoring interventions. It can be expected that the development of other contrast media for MRI and different modalities will continue in the near future.

### 3.2. Mass Spectrometry Imaging

Mass spectrometry imaging (MSI) is a powerful tool capable of imaging and profiling various molecules with high sensitivity, i.e., subcellular structures, and individual cells, without labeling in a single experiment, e.g., intracellular localization of pharmaceuticals. However, the disadvantage is that the use of MSI absolute quantification is usually not possible, as opposed to secondary ion mass spectrometry (SIMS), because of the diversity of factors that affect the intensities of ion signals recorded within the region of interest.

SIMS operates in the MSI mode and can routinely achieve spatial resolutions at the sub-micrometer level. Therefore, it is a powerful tool that is often used in micro-bioanalytical investigations and drug distribution studies [97]. Due to this dynamic, SIMS was used for the quantitative mapping of boron directly at subcellular resolutions, allowing a successful evaluation of the effectiveness of various BNCT pharmaceuticals and comparison of boron concentration in subcellular regions [98].

Two directions of studies focused on the use of SIMS in BNCT can be distinguished: (I) microprobe methods combined with post-ionization laser techniques [54,99], and (II) use of the ion microscope technique by applying a high current primary beam O_2_^+^ and then using a position-sensitive detector that detects positive secondary ions [54,96,100].

Chandra et al., successfully performed many SIMS-based investigations and quantitative evaluations on boron neutron capture therapy drugs. The evaluation of free or loosely bound boron pools was performed in the cytoplasm and nucleus of cryogenically prepared cultured human glioblastoma multiforme cells exposed to BPA. Both evaluated boron agents delivered ~70% of the boron pool in bound and mobile form to the nucleus and cytoplasm [96,100].

Aldossari et al. [54] also conducted an application study for the localization and quantification of therapeutic levels of the BNCT agent L-para-(dihydroxyboryl)-phenylalanine (BPA) in a primary cell using a high-resolution dynamic SIMS instrument. Cell cultures were obtained from patients (humans) who suffered from glioblastoma multiforme tumors.

It is also worth noting that boron measurements at the subcellular level in the cytoplasm and nuclei samples—collected after fractionation of tumor cells—cannot also be made with high confidence by bulk methods of determination of boron concentration, which are vital to BNCT. Free and loosely bound boron pools would be lost (more likely) from their native subcellular locations, e.g., during the liquid centrifugation or in other steps of fractionation. Bulk techniques cannot also determine the increased accumulation of ^10^B within the cell nucleus [96].

### 3.3. Single-Photon Emission Computed Tomography and Prompt Gamma-Photon Detectors

A precise real-time measurement of the ^10^B spatial distribution in healthy and pathological tissues are required to take full advantage of BNCT selectivity, as it would improve the effectiveness of the BNCT, and therapy personalization for each patient. Many feasibility analyzes of a single-photon emission computed tomography (SPECT) instrument for quantifying the boron dose have been carried out over the years [101]. Some of them would provide only dosimetric data, such as the absolute number of BNCT reactions that occur within the measured region [102,103]. Subsequently, a modification of the BNCT SPECT [104] information was proposed that allows one to determine the boron concentration in real-time. It is based on the number of neutrons that pass through the patient, measured by taking advantage of the cadmium neutron capture ^113^Cd (*n*, γ)^114^Cd reaction occurring in the detector. One of the main advantages of measuring the 477 keV photon emitted after ^10^B capture reaction during deexcitation of the ^7^Li recoil nucleus (Figure 2) is imaging the same molecule as used for therapy, without the need to introduce and develop a specific tracer.

Prompt gamma-ray spectroscopy in BNCT is another method to detect the concentration of boron. It is a similar approach as applied in SPECT. Many different devices (e.g., CZT drift strip detectors [105], CdTe semiconductor detectors [106,107]; scintillator detectors [108]) have been proposed during numerous research studies to promote the clinical translation of this method. Tian et al. [39] proposed a dual prompt gamma detection method that could allow accurate three-dimensional determination and reconstruction of the boron concentration in vivo and the dose distribution in the region of interest (ROI) during BNCT. This method is based on the relationship between ^10^B (*n*, α)^7^Li and ^1^H (*n*, γ)^2^H reactions. However, there are still many technical challenges to be solved before implementing this method in clinical applications.

As pointed in the physical basis and dosimetry section, imaging 477 keV photon may serve also as a dosimetric tool. Unfortunately, there are also 2.2 MeV gammas originating from reactions with ^1^H [109]. In a model paper [110] Goodman et al., studied the dose delivered to the patient using BSH in BNCT. The study used only simulation for dose calculation. Verbakel [111] has proposed a gamma-telescope that allows dosimetry during treatment. In these applications, the dose is calculated from assumed RBE and based on monitoring or imaging of prompt gamma of 477 keV and is not an actual dosimetry. The micro-dosimetry for BNCT in Tissue Equivalent Proportional Counters was developed by Moro et al. [112,113].

## 4. Clinical Possibilities of BNCT and Future Perspectives

This section focuses on the potential clinical applicability of BNCT therapy, starting from recent technological advances in producing neutron beams with suitable energy and intensity, followed by the presentation of clinical trials already closed and others currently underway. Later, the singular aspects that indicate BNCT as an individualized and personalized approach to treating cancer are presented.

### 4.1. New Compact Linac-Based BNCT Neutron Sources

One of the reasons why BNCT is once again a topic of interest in RT, despite not being a new technique as discussed above, is precisely because of the recent generation of new compact devices that could treat patients in conditions similar to those of other particle accelerators in hospital settings.

For BNCT in boron-labelled tumor cells, an adequate thermal neutron field must be created. Therefore, a neutron source that meets the guidelines of the International Atomic Energy Agency (IAEA) is required [65]. This guide was specifically written for the application of BNCT for tumors in tissue deep inside the patient, such as brain cancer. Epithermal neutrons are considered the most effective under a recommended ratio regarding thermal neutrons, and with low enough levels of components in the flux of fast neutrons and gamma rays. This guideline was written for reactor neutron sources considered as the only facilities able to provide the beam performance mentioned above, which has slowed the possibilities of the clinical application of BNCT because of the difficulty of building a new reactor just for BNCT near a hospital. This IAEA document is currently undergoing an update process. Medical uses require lower costs and higher stability of therapeutic beams, and they only can be proportioned by an accelerator-based neutron source. Recent technological advances could again promote BNCT as a real clinical option in conditions not so different from heavy particle installations and raise similar expectations about the benefits over conventional radiation therapy [114].

Figure 5 represents a comparison of the facilities size, between a compact accelerator of protons and another one based on the BNCT of the last generation. The world’s first accelerator-based system for clinical BNCT irradiation (C-BENS) with the cyclotron of Sumitomo Heavy Industries has been developed at the Kyoto Research Reactor Institute [115]. Within Europe, Neutron Therapeutics designed one of the first commercial accelerator-based Boron Neutron Capture Therapy Platforms and installed it in the Helsinki University Hospital. This facility is the base for the process of obtaining the European CE safety certification for medical devices, necessary for their commercialization. This process is currently ongoing there. TAE Life Sciences (TLS) is a company for biologically targeted radiation therapy developing a breakthrough accelerator-based BNCT system, which has formalized an agreement with the National Center of Oncological Hadron-therapy (CNAO) in Italy to provide its Alphabeam™ Neutron System, which integrates RayStation of RaySearch Laboratory, the more extended software for treatment planning in RT. The latter is a clear signal of the new interest in BNCT, despite being a fairly old technique, due to the recent generation of these new compact devices that could treat patients in conditions similar to those of other particle accelerators in hospital settings.

### 4.2. BNCT Clinical Trials

Until recently, the value of BNCT was largely restricted and the number of patients treated with BNCT was limited because treatment could only be performed in nuclear research reactors—the only neutron source at the time—as previously discussed in Section 4.1. Furthermore, most of the clinical trials conducted were carried out in facilities at nuclear reactor sources. With the improvement of neutron beam-generating instruments, BNCT would be able to improve the robustness of the clinical trial results by increasing the number of patients included in these studies. BNCT has been the subject of several clinical trials over recent decades.

The first clinical trials date back to the 1950s [23,24] using first-generation compounds with poor biodistribution, which was the main cause that led to their failure, as previously discussed [118,119,120]. Some of the first reported clinical trials using second-generation boron carriers date back to the late 1980s/early 1990s [121].

Several locations have been key in the development of BNCT and its clinical trials, being one of them the Finnish Research Reactor (FiR 1 reactor), where some clinical trials started at 1999. In 2003, Joensuu et al., summarized the ongoing clinical trials at these facilities: one trial for glioblastoma patients who had not undergone surgery or radiotherapy before BNCT, and another for recurring or progressing glioblastoma patients who had previously undergone surgery or conventional cranial radiotherapy. Both phase I/II clinical trials concluded that BNCT was well tolerated and established the foundations for fructose-BPA-based BNCT applications in the clinic [122]. Later, in 2007, Kankaanranta et al., published the results of another Finnish trial that evaluated BNCT as a treatment for locally recurrent head and neck cancer in another phase I/II clinical trial with two irradiation sessions. The results showed that BNCT was effective and safe to use in patients with previously irradiated head and neck cancer who had recurrences [123]. Almost five years later, in early 2012, a final analysis revealed that most of the patients were BNCT respondents, some of them were progression-free for sustained periods, and only one patient of the cohort progressed in their disease stage. They also reported acceptable toxicity levels [124]. Additionally, in 2011, Kankaanranta et al., presented the results of a clinical trial evaluating BNCT in malignant gliomas resistant to surgery and conventional radiotherapy. The main objective of this trial was to determine the biosafety of different doses of BPA ranging from 290 mg/kg to 450 mg/kg, which was determined as the maximum tolerated dose in which some adverse effects appeared. Furthermore, they estimated that irradiation doses were higher in patients administered with more than 290 mg/kg of fructose-BPA compared to those administered only 290 mg/kg. Therefore, they concluded that BPA-based BNCT with doses of up to 400 mg/kg was a feasible treatment for malignant recurrent gliomas [125]. The data from the patients in this study was used afterwards to formulate and train a pharmacokinetic model of boron biodistribution in glioma patients [126].

In the UK, in 2009, Cruickshank et al., reported preliminary data from a clinical trial started in 2007 to evaluate BPA pharmacokinetics for patients with glioma. More specifically, they worked on the location of the intravenous infusion and the use of mannitol as a controlled blood–brain barrier disrupter, and its effect on boron biodistribution. However, this trial was terminated and no further information was found. The only information shared was from a single patient infused without mannitol, whose biodistribution data were consistent with data from Finnish trials [127].

In late 2016, Yong et al., reported the outcome of their first case in the clinical trial as the preliminary results of the In-Hospital Neutron Irradiator for treating malignant melanoma with BNCT in China. In this phase I/II clinical trial, BPA-fructose was used at a dose of 350 mg/kg infused intravenously for 90 min and the concentration of circulating boron was measured over time by performing several blood extractions of the patient. Irradiation was carried out successfully and the patient showed a complete response to BNCT without late radiation injury and only some minor grade 2 acute radiation damage that resolved after proper treatment [128].

In Taiwan, a phase I to II BNCT clinical trial was conducted between 2010 and 2013 for patients with recurrent head and neck cancer. They were treated with a two-session BNCT treatment using fructose BPA, and reported a wide range of responses, including a complete response in half of the patients, and partial responses with various reactions, some of which were disease-free for more than 50 months after BNCT [129]. A second clinical trial, also phase I/II, was started in 2014 combining BNCT with IGRT and IMRT to try to reduce re-recurrences [130]. As of 2019, Lee et al., provided evidence suggesting that the single-session dose distribution in terms of homogeneity and conformity might improve when combined with multifraction IMRT [131]. Additionally, in Taiwan, BNCT was evaluated as a salvage therapy for malignant brain tumors, with preliminary results published in 2020 showing significant effects on maintaining a good quality of life and possibly prolonging a patient’s survival [132]. Their latest results were published in 2021, showing no severe adverse effects, a possible increase in survival, and also establishing some key parameters and recommendations that patients should meet to undergo BNCT or at least the conditions under which BNCT showed better results [133].

At the time of writing this review, there is one open and recruiting BNCT-related clinical trial listed in the ClinicalTrials database with reference NCT04293289. This study based in Japan is a phase I clinical trial of BNCT for malignant melanoma and angiosarcoma using an experimental CICS-1 BNCT accelerator and SPM-011 (borofalan) intravenously at 200 mg/kg/h for 2 h before neutron irradiation and a continuous infusion at 100 mg/kg/hour during irradiation. This trial started in late 2019 and is expected to be finished by the end of 2022.

Although these trials have been steadily increasing, the lack of phase III trials comparing the effectiveness of BNCT against conventional radiation treatment is a flaw to overcome in the coming years, as well as fully exploring the features provided by BNCT towards a more patient-specific approach.

### 4.3. BNCT and Personalized Radiotherapy

RT in general is searching for new methodologies for rational patient treatment based on molecular knowledge and involving imaging in radiobiology to assess the therapeutic outcome of targeted drugs in combination with radiation, in addition to the toxicities that limit the escalation of prescribed dose of the treatments.

As discussed previously, in recent decades, radiotherapy has experienced significant advances, mainly due to technological progress. These advances led to prominent levels of precision in delivering to the tumor volume a prescription dose based on population evidence. However, new research lines are trying to find treatment plan and a prescribed dose specific for each patient once delivery precision has been assured.

In this sense, it is necessary to characterize the biological reality of the tumor, and, hence, the latest developments in radiotherapy are related to its personalization at a molecular level. At this point, both omic techniques and functional imaging are the key approaches used to make even more personalized radiotherapy, integrating patient-specific molecular data with population-based data, such as clinical endpoints and toxicity constraints [134]. Examples of such personalization are the use of radiotracers that label hypoxia levels at a treatment planning level, contouring, and prescribing doses based on zones with different proliferation levels, and even using RT as an immunotherapy adjuvant for oligometastatic disease [135,136,137,138,139]. The personalization level has been brought even to prescribing doses to different tumoral regions identified in functional imaging that obey different radiation resistance levels and/or to the true pathoanatomical borders of the tumor. This specific dose prescription based on the heterogeneity provided by the molecular image has been proposed even up to individual voxels corresponding to the tumor volume after the appropriate segmentation process in the image [140]. This approach has been named dose painting [141] and is being slowly implemented in treatment planning because harmonization protocols must still be established previously between a larger number of image devices worldwide, specifically PET/CT [142], enough to achieve common quantitative metrics from a larger cohort, thus being able to confirm the robust correlation between the gray level in the image and the biological information sensible to the prescription dose decision in each case.

As Ree et al. argued, considering these biological and molecular characteristics can contribute to selective sensitization of the tumor for optimization of tumor control, reducing normal tissue injury from radiation damage, and even modulating the immune system for better management of locally advanced, and possibly also metastatic disease [134].

The same principles apply to BNCT, where direct personalization could be achieved through selective molecular imaging based on boron biodistribution for subsequent highly precise targeted radiotherapy planning, as discussed in Section 2. Nevertheless, approaches based on high throughput omics will also be desirable. As Sauerwein et al., discussed in early 2021, BNCT personalization is based on the ^10^B targeting uptake mechanism, tunable using different vectors based on previously determined molecular characteristics of the tumor. This uptake would affect the radiation dose delivered at the cellular and even subcellular levels, but more importantly, this boron uptake and biodistribution will also depend on the characteristics of each patient from a pharmacogenetic point of view [36]. Therefore, several models have been developed to evaluate and predict the pharmacokinetics and irradiation effect of BNCT using different molecular ligands in boron transporters, such as the in vitro model proposed by Ishiyama et al. [136] using three-dimensional tumoral human cell cultures.

Regardless, focusing on proper molecular profiling using high throughput approaches is a fundamental step that needs to be tackled in the near future for more complete personalization of BNCT. These studies should not only be limited to genomics or transcriptomics, but also include proteomics or metabolomics to have a clearer picture of the problem, identify and classify the different stages of the disease more effectively, and avoid the false static and limited view that can provide genetic approaches.

Experimental studies also evaluated the relationship between tumor temperature and the response to BNCT application. The main objective was to measure temperature before irradiation and to check if it could be used as a predictive indicator of boron tumor uptake, so BNCT could be personalized and optimized for each individual patient by adjusting neutron fluence [143].

Few omic approaches for BNCT have been reported. Some of the first procedures related to BNCT proteomics were those discussed by Mauri and Basilico, in 2012, where the focus of the study was to determine the interaction of boron transporters with endogenous proteins, using techniques, such as bidimensional gel electrophoresis and multidimensional protein identification [144].

Later, in 2015, Sato et al., studied the cellular response to BNCT in SAS human squamous cell carcinoma, with and without previous incubation with BPA. Their findings showed that the proteome presented alterations related to processes, such as DNA repair and RNA processing, as well as changes in proteins located in the cellular compartment of the endoplasmic reticulum. Thus, such proteins could be involved in the early response to BNCT. More precisely, they found that lymphoid-restricted membrane proteins were induced after BNCT and may be related to BNCT-induced cell death [83].

In 2019, Ferrari et al. [84] reported a proof-of-concept study in which urinary samples were used to characterize the proteome of patients with squamous cell head and neck cancer and thyroid cancer in a BNCT clinical trial. Samples were taken before and after the administration of the boron agents. Several candidate biomarkers were found and changes in the proteome were detected after boron infusion, more precisely the reduction in the expression levels of three inflammation-related molecules.

In any case, as Keener pointed out in late 2020, whether through genetics or advanced imaging, the field of radiation oncology is slowly but steadily adopting the principles of personalized medicine [145] and BNCT shows a potential higher efficiency for clinical implementation since it provides cellular targeting of energy transfer radiation and, therefore, can provide more useful previous genetic and biological information.

## 5. Conclusions

Boron Neutron Capture Therapy incorporates the specific principles of some chemotherapies and targeted therapies into the precise location principles of conventional radiotherapy. In this way, BNCT could share the typical methodologies used in personalized medicine. Since some types of cancer, such as glioblastoma, remain exceptionally resistant to all current forms of therapy, such as chemotherapy, surgery, radiotherapy, and immunotherapy, BNCT is a promising option for these types of tumors. However, some critical issues must be resolved if BNCT is to become a better and more valuable cancer treatment. Since controlled intracellular targeting is of great importance in inducing the cell-killing activity of BNCT due to specific cell death pathways, such targeting should be further assessed, together with the conduct of adequate research and clinical trials to determine the most profitable and promising routes in drug delivery systems. Activation of the DNA response, such as damage and repair mechanisms of complex double-strand DNA break activated by a mixed neutron-gamma beam, has been poorly studied and, therefore, it is not fully determined. A deeper understanding of how cells preferentially select specific DNA damage responses generated by high-LET and mixed radiation, and detailed mechanisms of enhanced necrosis due to ADP depletion, can lead to improved therapeutic efficiency in BNCT. Individual tumor cell quantification of bound and free pools (net cellular content) of ^10^B needs to be further addressed, as it remains challenging (owing to insufficient spatial resolution) with clinically applicable techniques. Studies designed to test and improve boron detection methods could reduce detection limits and identify accumulation regions in tumor cells and normal tissues with greater precision. Further research on mechanisms for detecting the distribution of prompt gamma rays that arise during BNCT could also be profitable. However, the ideal dose paradigm for BNCT, the real-time measurement of the distributions of reactions, such as ^10^B(n,α) ^7^Li and ^14^N(n,p)^14^C, and the quantitative mapping of the boron concentration in the body have not yet been determined. Finally, the applicability of the BNCT boron-containing drug used in the individual patient must be evaluated for effective treatment, not only by PET and other individual imaging methods, but also by proteomics and gene expression methods, as is beginning to be performed for BPA (Borofalan).

## Figures and Tables

**Figure 1 cancers-14-02865-f001:**
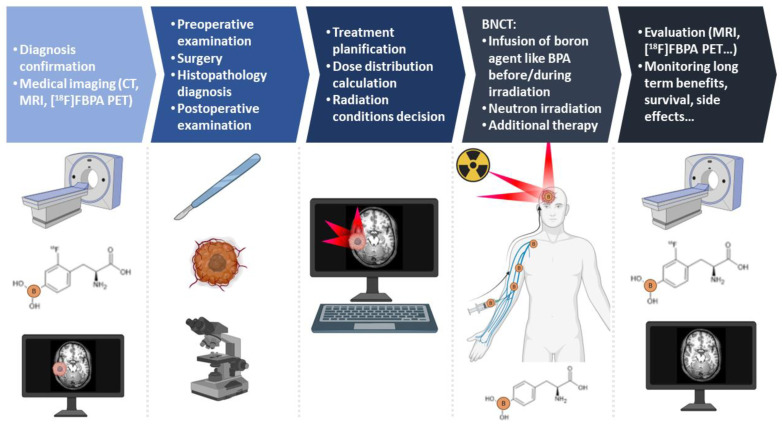
Standard BNCT procedure. From prior medical examination and diagnosis confirmation with the use of [^18^F]FBPA PET, through Boron Neutron Capture therapy, to post-therapy monitoring and evaluation.

**Figure 2 cancers-14-02865-f002:**
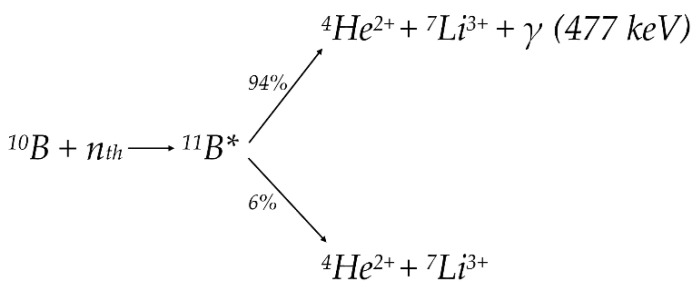
^10^B (*n*, α)^7^Li reaction.

**Figure 3 cancers-14-02865-f003:**
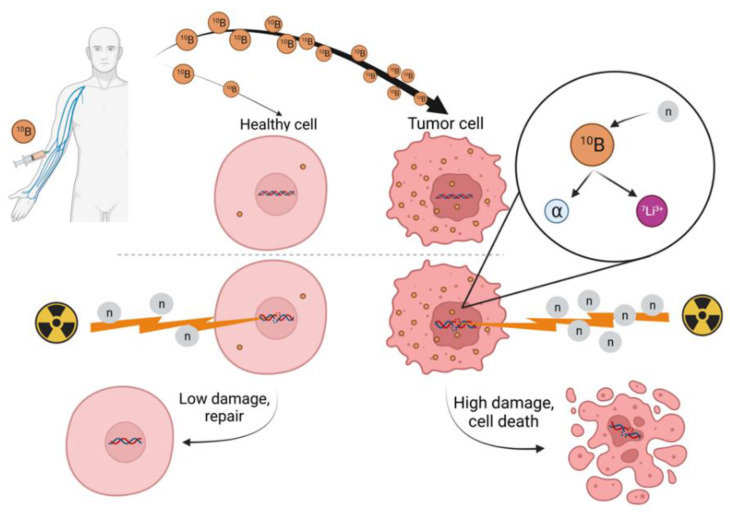
Schematic representation of BNCT principles of action at a cellular level. Boron accumulation is favored in tumoral cells (thick upper arrow) in comparison with normal cells (thin upper arrow).

**Figure 4 cancers-14-02865-f004:**
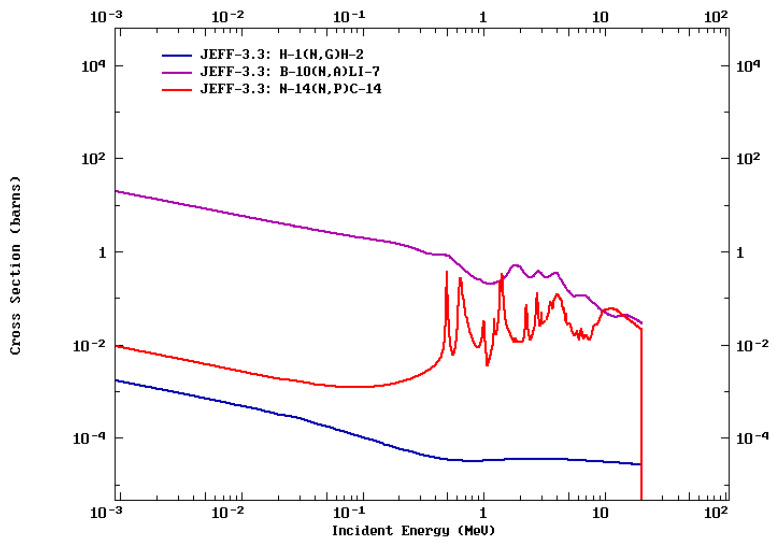
Cross-sections of reactions that take place because of neutron collisions. ^1^H (*n*, γ)^2^H (blue), ^10^B (*n*, α)^7^Li (magenta), ^14^N (*n, p*)^14^C (red). The figure was plotted using data from ENDF [64].

**Figure 5 cancers-14-02865-f005:**
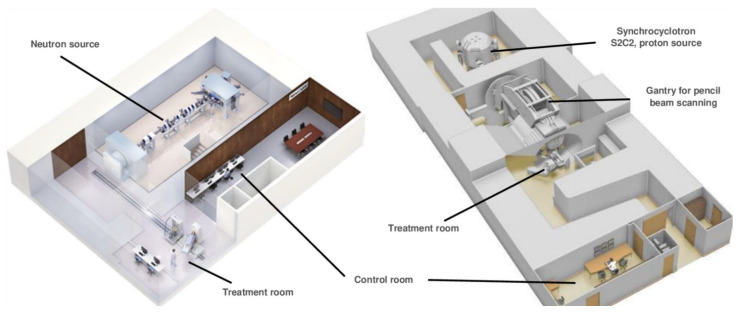
On the left side, the Sumitomo BNCT System NeuCure was recently approved by the Japanese government and is running in two hospitals from early 2020. Image adapted from [116]. On the right side, a compact accelerator of Proteus^®^ONE protons, installed at Willis-Knighton Center in Los Angeles, USA, clinically used from 2014. Image adapted from [117].

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
