# Peer review of "Clinical Viability of Boron Neutron Capture Therapy for Personalized Radiation Treatment"

_cancers, 2022, doi:10.3390/cancers14122865_

Round 1
Reviewer 1 Report
The Authors present a valid review on the Clinical viability of Boron Neutron Capture Therapy for personalized radiation treatment, which is arousing new growing interest thanks also to the improvement of radiation sources and the design of new carriers.
The only advice I give to the authors before publishing the work is to implement the resolution of the figures and to insert at least one new figure on the mechanisms of the effects of BNCT at the cellular level updated to date.
English could be slightly implemented,
I congratulate the authors and I hope that it will be accepted on CANCERS MDPI after such minor revisions.
Author Response
Point-by-point response:
The Authors present a valid review on the Clinical viability of Boron Neutron Capture Therapy for personalized radiation treatment, which is arousing new growing interest thanks also to the improvement of radiation sources and the design of new carriers.
The only advice I give to the authors before publishing the work is to implement the resolution of the figures and to insert at least one new figure on the mechanisms of the effects of BNCT at the cellular level updated to date.
Thank you for your comments. We have improved the resolution and quality of the previously existing figures. We have also curated a new figure on the schematic mechanisms of BNCT at the cellular level.
English could be slightly implemented,
We have now checked English, and used Writefull software to improve it. We have also submitted the text to a native English speaker.
I congratulate the authors and I hope that it will be accepted on CANCERS MDPI after such minor revisions.
Thank you very much. Honestly, we believe that the manuscript has now improved after addressing the reviewers' suggestions.
Reviewer 2 Report
The authors well addressed the recent BNCT protocol and described why BNCT can be regarded as personalized radiation therapy in this manuscript, but here is one issue that should be addressed before publication.
1. In the 2.2. section (Mechanisms of Cell Death in BNCT), the authors mentioned that the Monte Carlo technique is the most advanced method of calculating fluxes and doses in complex geometries. I suggest that the authors briefly discussed the dose simulation software and methods and how these techniques facilitate clinical personalized treatment planning in this section.
Author Response
Point-by-point response:
The authors well addressed the recent BNCT protocol and described why BNCT can be regarded as personalized radiation therapy in this manuscript, but here is one issue that should be addressed before publication.
- In the 2.2. section (Mechanisms of Cell Death in BNCT), the authors mentioned that the Monte Carlo technique is the most advanced method of calculating fluxes and doses in complex geometries. I suggest that the authors briefly discussed the dose simulation software and methods and how these techniques facilitate clinical personalized treatment planning in this section.
Thank you for your comments. Following suggestions of referees, we added a new section (2.3. Physical basis and dosimetry of BNCT) where we discussed Monte Carlo methodology linked to BNCT and we added several references about it.
Thank you very much. Honestly, we believe that the manuscript has now improved after addressing the reviewers' suggestions.
Reviewer 3 Report
The authors write a review on BNCT with the intention of providing an overview of this emerging targeted therapy for the personalized radiation treatment of cancer. In the review they describe several aspects closely related to BNCT: the fundamental principles, recent advances, and future directions.
I personally consider the paper interesting but recently, other reviews have already been published on the subject 1) Dymova et al Boron neutron capture therapy: Current status and future perspectives. Cancer Communications 2020 https://doi.org/10.1002/cac2.12089
2) Malouff et al Boron Neutron Capture Therapy: A Review of Clinical Applications. Front Oncol. 2021 doi: 10.3389/fonc.2021.601820. None, however, deals in detail with the dosimetric part image tools and clinical trails as is done by the authors. My advice is to look more into the first topic considering that it is of particular importance for the success of the treatment, as indicated at various points in the paper by the authors themselves. In particular, I would write a section on dosimetry dividing it into physical dosimetry (simulations and physical approaches for dose calculation) and biological dosimetry (for dose calculation and improving treatment outcomes)
Here are some suggestions to make the paper clearer and some editing corrections.
Line 81-82: to delete “The goal of RT is to achieve more accurate and efficient dose delivery to organs and tissues” because it is a repeated concept in both the first sentence and the next sentence
Line 102: In my opinion, at the end of the sentence “BNCT is a binary treatment method based on the combination of two agents, 10B and epithermal neutrons, which exploits the high-LET characteristics of the mixture of fractionation components” the whole pargaraph in which the two types of neutrons used (line 562 and later) should be put here. Among other things, the difference is not clarified and it is not explained why better the epithermal ones than the thermal ones.
Line 158: to use the acronym BNTC
Line 195: to delete “in the range +1...+2” the ellipsis
Line 209-210: put at least one reference
Line 232: before section 2.2 Mechanisms of Cell Death in BNCT I would insert section 4.3. BNCT and personalized radiotherapy. In my opinion the reading and understanding of the paper becomes clearer.
Line 233: section 2.2. Mechanisms of Cell Death in BNCT. In my opinion, the first paragraph is unclearly written with some inaccuracies. I recommend revising it and suggest here “In BNCT, cells are damaged mainly by alpha particles or the 7Li ion, as they can cause various DNA lesions (i.e. DNA damage in clusters or multiple local damage sites) along their path, resulting in genome instability. In addition to DNA, macromolecules can also be damaged, resulting in modulation of their functions [36]. When these particles pass through a cell, their path is short (α < 10μm and 7Li < 5μm) [37], so their kinetic energy is released within the target cell, whose diameter is usually ~10μm. Therefore, it does not affect surrounding healthy cells.
Line 245: Unclearly written sentence. I suggest: Radioinduced damage can be produced through two types of action: direct and indirect
Line 252: Delete the sentence “These free radicals are formed in irradiated body tissue and blood cells” or write it more clearly. As it is written in the text it is misleading. Free radicals are produced at all sites affected by radiation
Line 256: dissolve acronym LET
Line 256-275: All of this part, with appropriate changes and supplements as the authors consider appropriate should be put in the dosimetry paragraph.
Line 274-275: The phrase “DNA damage increases together with radiation LET [42], and the higher the LET, the higher the relative biological effectiveness (RBE)" is perhaps a little too strong. The RBE LET trend is complex, and the RBE increase as function of LET also depends on the endpoint considered. In some cases the RBE may increase with increasing LET and then decline above certain LET values. Takatsuji et al. doi: 10.1269/jrr.40.59 discusse the relationship between LET and RBE for chromosomal aberrations and cell death: at low doses the RBE increase with increasing LET, then RBE value peaks at an LET of about 100 keV µm-1, and finally the RBE decreases as LET increases further.
Line 301: dissolve acronym ncRNA
Line 310: dissolve acronym HRR NHEJ
Line 323: missing parentheses and the acronym RLA
Line 379: to use the acronym BNTC
Line 395-403: also this part should go in the physical dosimetry paragraph because it influences the dose calculation.
Line 413: to use the acronym PET
Line 423: write in vivo in italics
Line 448: to use the acronym MRI
Line 449: write in vivo in italics
Line 485: Reference 59 refers to item II) and is then described in the last paragraph. A bibliographical reference of the first item is missing. The authors refer to refs. 74 and 77? If so, I recommend adding these references after point I)
Line 502: missing space “years[78]”
Line 508-511: revise the paragraph. A piece of sentence is repeated
Line 511: missing space “tracer.Li”
Line 523: This paragraph 3.4. Other molecular imaging tools in BNCT seems unnecessary to me because some concepts could be added to those written in the previous parts. Example [18F]FBPA and [11C]-methionine are used with PET and so this part could go in the paragraph 3.1. Positron emission tomography and magnetic resonance imaging. Also, PHITS in this paragraph has nothing to do with it and instead could go in the dosimetry paragraph.
Line 596: delete boron neutron capture therapy
Line 604: 4.2. BNCT clinical trials. Write the trails in chronological order
Ref 74: the authors' names should all be written in lower case letters
Author Response
Point-by-point response:
The authors write a review on BNCT with the intention of providing an overview of this emerging targeted therapy for the personalized radiation treatment of cancer. In the review they describe several aspects closely related to BNCT: the fundamental principles, recent advances, and future directions.
I personally consider the paper interesting but recently, other reviews have already been published on the subject 1) Dymova et al Boron neutron capture therapy: Current status and future perspectives. Cancer Communications 2020 https://doi.org/10.1002/cac2.12089; 2) Malouff et al Boron Neutron Capture Therapy: A Review of Clinical Applications. Front Oncol. 2021 doi: 10.3389/fonc.2021.601820. None, however, deals in detail with the dosimetric part image tools and clinical trails as is done by the authors. My advice is to look more into the first topic considering that it is of particular importance for the success of the treatment, as indicated at various points in the paper by the authors themselves. In particular, I would write a section on dosimetry dividing it into physical dosimetry (simulations and physical approaches for dose calculation) and biological dosimetry (for dose calculation and improving treatment outcomes)
Thank you very much for your comments. We are glad you find this work interesting. We corrected the grammar thanks to a native expert in the field, and specialized software was applied to the text.
Attending your comments above, we have created a new section regarding physical bases and dosimetry, and slightly modified the biological dosimetry section. Now, we think the manuscript is going deeper into these issues.
Here are some suggestions to make the paper clearer and some editing corrections.
Line 81-82: to delete “The goal of RT is to achieve more accurate and efficient dose delivery to organs and tissues” because it is a repeated concept in both the first sentence and the next sentence
This has been corrected in the new version.
Line 102: In my opinion, at the end of the sentence “BNCT is a binary treatment method based on the combination of two agents, 10B and epithermal neutrons, which exploits the high-LET characteristics of the mixture of fractionation components” the whole pargaraph in which the two types of neutrons used (line 562 and later) should be put here. Among other things, the difference is not clarified and it is not explained why better the epithermal ones than the thermal ones.
We thank you for your suggestion. We have modified the text in the new version accordingly, to make it clearer.
Line 158: to use the acronym BNTC
This has been addressed in the new version.
Line 195: to delete “in the range +1...+2” the ellipsis
This has been addressed and the range has been stated more precisely, with a newly added reference.
Line 209-210: put at least one reference
Two references have been added regarding some clinical trials using such compounds. Additionally, although those trials date back to the starting point of BNCT which is not the point of the discussed clinical application, some information about them has also been added to the “Clinical Trials” section.
Line 232: before section 2.2 Mechanisms of Cell Death in BNCT I would insert section 4.3. BNCT and personalized radiotherapy. In my opinion the reading and understanding of the paper becomes clearer.
We understand the reasons of the referee to change the location of that section, because it is a doubt for ourselves. Finally, we decided to add some sentences at the end of section 2.1 for helping the understanding of the manuscript. While we already considered including BNCT and personalized radiotherapy as a section after 2.1., we placed it in the last part as it focuses on the future perspectives of BNCT, being the personalized approach one of those.
Line 233: section 2.2. Mechanisms of Cell Death in BNCT. In my opinion, the first paragraph is unclearly written with some inaccuracies. I recommend revising it and suggest here “In BNCT, cells are damaged mainly by alpha particles or the 7Li ion, as they can cause various DNA lesions (i.e. DNA damage in clusters or multiple local damage sites) along their path, resulting in genome instability. In addition to DNA, macromolecules can also be damaged, resulting in modulation of their functions [36]. When these particles pass through a cell, their path is short (α < 10μm and 7Li < 5μm) [37], so their kinetic energy is released within the target cell, whose diameter is usually ~10μm. Therefore, it does not affect surrounding healthy cells.
We thank you for your suggestion. We have rephrased the text.
Line 245: Unclearly written sentence. I suggest: Radioinduced damage can be produced through two types of action: direct and indirect
The paragraph has been rewritten, including your suggestion.
Line 252: Delete the sentence “These free radicals are formed in irradiated body tissue and blood cells” or write it more clearly. As it is written in the text it is misleading. Free radicals are produced at all sites affected by radiation
The sentence has been deleted in the new version.
Line 256: dissolve acronym LET
The acronym was not dissolved before in the text. Now, it has been dissolved in the place it takes its first appearance.
Line 256-275: All of this part, with appropriate changes and supplements as the authors consider appropriate should be put in the dosimetry paragraph.
Thank you for your comment, we have included this in the new “physical dosimetry” section that we have added following the suggestion of the referees.
Line 274-275: The phrase “DNA damage increases together with radiation LET [42], and the higher the LET, the higher the relative biological effectiveness (RBE)" is perhaps a little too strong. The RBE LET trend is complex, and the RBE increase as function of LET also depends on the endpoint considered. In some cases the RBE may increase with increasing LET and then decline above certain LET values. Takatsuji et al. doi: 10.1269/jrr.40.59 discusse the relationship between LET and RBE for chromosomal aberrations and cell death: at low doses the RBE increase with increasing LET, then RBE value peaks at an LET of about 100 keV µm-1, and finally the RBE decreases as LET increases further.
Thanks, it is true that in the first version, we generalized on this concept. We have included the provided reference and made some clarifications in the newest version.
Line 301: dissolve acronym ncRNA
The acronym has been removed, as this term is only mentioned once. Now it is mentioned as non-coding RNA
Line 310: dissolve acronym HRR NHEJ
They have been dissolved in the new version.
Line 323: missing parentheses and the acronym RLA
Some rephrasing has been done in order to clarify the text. However, RLA has been kept that way, as is not an acronym but the sequence of the peptide (RLA: Arginine-Leucine-Alanine).
Line 379: to use the acronym BNTC
This has been addressed in the new version.
Line 395-403: also this part should go in the physical dosimetry paragraph because it influences the dose calculation.
This part has been moved to the new physical dosimetry part.
Line 413: to use the acronym PET
This has been addressed in the new version.
Line 423: write in vivo in italics
This has been addressed in the new version.
Line 448: to use the acronym MRI
This has been addressed in the new version.
Line 449: write in vivo in italics
This has been addressed in the new version.
Line 485: Reference 59 refers to item II) and is then described in the last paragraph. A bibliographical reference of the first item is missing. The authors refer to refs. 74 and 77? If so, I recommend adding these references after point I)
Referencing on this paragraph has been fixed accordingly.
Line 502: missing space “years[78]”
This has been corrected in the new version.
Line 508-511: revise the paragraph. A piece of sentence is repeated
Thank you for pointing out this mistake. We have already corrected it.
Line 511: missing space “tracer.Li”
This was provoked due to the error pointed out previously resulting in a duplication of the sentence, and thus has been corrected in the new version.
Line 523: This paragraph 3.4. Other molecular imaging tools in BNCT seems unnecessary to me because some concepts could be added to those written in the previous parts. Example [18F]FBPA and [11C]-methionine are used with PET and so this part could go in the paragraph 3.1. Positron emission tomography and magnetic resonance imaging. Also, PHITS in this paragraph has nothing to do with it and instead could go in the dosimetry paragraph.
We completely agree, this makes the manuscript clearer and improves the reading flow. We thank you for your suggestion and confirm that have removed section 3.4, including the content in the rest of the sections.
Line 596: delete boron neutron capture therapy
This has been addressed in the new version.
Line 604: 4.2. BNCT clinical trials. Write the trails in chronological order
We have restructured some of the parts of this section. However, we think that a complete strict chronological order could lead to some confusions, as some clinical trials report results several times, and some others started as a continuation of previous ones. That is why, while we have tried to maintain a fully chronological order, there are some ruptures on the timeline to consider different places and non-related studies.
Ref 74: the authors' names should all be written in lower case letters
This has been corrected in the newest version.
Thank you very much. Honestly, we believe that the manuscript has now improved after addressing the reviewers' suggestions.
Round 2
Reviewer 3 Report
I thank the authors for accepting the suggestions and making the required corrections. In my opinion, the article has now improved, more complete and suitable for publication.